# COVID-19 hospitalisations and all-cause mortality by risk group in Finland

**Milla Summanen**[1], **Mikko Kosunen**[1], **Ville Kainu**[1], **Anniina Cansel**[2]*,
**Severi Niskanen**[2], **Lalli Nurmi**[2], **Riikka-Leena Leskelä**[2], **Outi Isomeri**[2]

**1** Pfizer Biopharma Group, Pfizer Oy, Helsinki, Finland, **2** NHG Finland, Nordic Healthcare Group, Helsinki, Finland

☯ These authors contributed equally to this work.
* anniina.cansel@nhg.fi

**Data Availability Statement:** All relevant data are within the paper and its Supporting Information files. Aggregate level data was used in the study. The non-aggregate level data is available from

## Abstract

Ever since COVID-19 was announced as a global pandemic in March 2020, healthcare systems around the world have struggled with the burden of the disease. Vaccinations and other preventive measures have decreased this burden, but severe forms of COVID-19 leading to hospitalizations and even deaths still effect certain risk groups, such as the elderly and patients with multiple comorbidities. The objective of this retrospective observational study was to identify which risk groups are at the highest risk for a severe COVID-19 infection in Finland using national registry data ranging from January 2021 to June 2022. The data was analysed in three time periods, enabling comparisons in high-risk groups between epidemiological waves caused by different variants of SARS-CoV-2. The summary level data were stratified according to predefined groups based on two criteria: age (≥18 years, 18–59 years, and ≥60 years) and risk group. The results include analysis of infection hospitalisation rate (IHR), case fatality rate (CFR) and average length of stay (LOS) in both primary and specialty care for each risk group and age group. Our results confirm that despite the decrease in COVID-19 hospitalisations and deaths observed during the study period, a significant proportion of patients are still hospitalised, and deaths occur especially in the 60+ population. Also, even though the average length of stay of hospitalised COVID-19 patients has decreased, it is still long compared to specialty care hospitalisations in general. Old age is a significant risk factor for severe COVID-19 in all patient groups and certain risk factors such as chronic kidney disease clearly increase the risk for severe COVID-19 outcomes. Early treatment should be considered with a low threshold for risk group patients and for elderly patients in order to avoid severe disease courses, and to ease the burden on hospitals where resources are currently very strained.

## Introduction

World Health Organisation declared COVID-19 as a global pandemic in March 2020 [1]. Since then, governments and healthcare systems have been pushed to the limits in the fight against it.

Finnish Institute for Health and Welfare (THL, PO BOX 30, FI-00271 HELSINKI, +358 29 524 6000).

**Funding:** The study was funded by Pfizer Oy. AC, SN, LN, R-LL and OI are employees of Nordic Healthcare Group, which received funding from Pfizer Oy in connection with the development of this manuscript. The funders participated in study design, analysis, decision to publish and preparation of the manuscript. The funders had no role in data collection.

**Competing interests:** The authors have declared that no competing interests exist.

While most people infected with COVID-19 have mild-to-moderate symptoms, a significant proportion may still experience severe symptoms leading to hospitalization and death [2–4].

The severe form of COVID-19 mostly affects certain risk groups; however, the disease can affect anyone regardless of age or pre-existing conditions [2]. A body of evidence has shown that people with pre-existing conditions, such as kidney, cardiovascular, chronic respiratory diseases, or diabetes are more likely to develop a severe form of COVID-19 that may lead to death [2–5]. Another important risk factor for severe COVID-19 is age [3, 4, 6, 7], and for example a previous study from Finland has shown that age is the most important risk factor for COVID-19 mortality [8]. Patients at increased risk of severe COVID-19 due to underlying medical conditions or age, i.e., risk group patients, tend to use more healthcare services and resources due to COVID-19, such as inpatient services in both primary and specialty care, compared to non-risk group patients [7], increasing healthcare costs and the burden on the healthcare system. This burden can be reduced by preventing disease transmission for example by physical distancing, good hand hygiene, and the use of face masks, and by vaccinations, but vaccinations cannot completely eliminate the risk of severe symptoms or transmission. These measures have been recommended especially for people at high risk of severe COVID-19 [9]. Severe forms of COVID-19 can also be prevented by treatments given in the early stages of a COVID-19 infection when the symptoms are still mild or moderate. However, the treatments are not always effective and should be used in combination with preventive measures and vaccinations.

Although age and certain medical conditions have been shown to increase the risk of severe COVID-19 as described above, more data is needed on whether these risk groups have remained the same throughout the pandemic and during epidemiological waves caused by different variants of the virus. Particularly, more data on which risk groups have the highest risk of severe COVID-19 during the omicron-dominant period would be useful for deciding which risk group patients should be eligible for early COVID-19 treatment options. Therefore, the main goal of this study was to identify which risk groups are at the highest risk of having a severe COVID-19 infection i.e., the likelihood of a COVID-19 infection leading to hospitalisation (either in primary care or in specialty care), and / or death in Finland at different time periods. Additionally, this study demonstrates the average healthcare resource use of these patients due to a COVID-19 infection. This retrospective observational study was conducted with Finnish registry data ranging from January 2021 to June 2022.

## Materials and methods

This study used aggregate data obtained from the Finnish Institute for Health and Welfare's (THL) registers: Care Register for Health Care (Hilmo), Register of Primary Health Care (Avohilmo), and Finnish National Infectious Diseases Register (TTR). The data from these registers was combined for this study on a patient level through unique patient identifiers and aggregated on a risk or age group level to get an overview of COVID-19 infections and hospitalisations related to COVID-19. Data was collected from January 2021 to June 2022 in six-month periods (1.1.-30.6.2021, 1.7.-31.12.2021, and 1.1.-30.6.2022). Data included four datasets: the number of infections, the number of hospitalisations and number of days spent in hospital care (separately for primary care and for specialty care), and the number of deaths. In Finland, patients are admitted to primary care wards and are under general practitioner surveillance if specialist medical care is not required, but the patient is not well enough to be released home.

The number of infections were defined as either a COVID-19 infection in the TTR or a COVID-19 diagnosis (ICD-10 code U07.1) in Hilmo. Data from Hilmo was used to identify patients for analysis, and any differences between TTR and Hilmo patients are reported in

supplementary materials (S1 and S2 Tables). Deaths were considered COVID-19 -related if they occurred within 30 days of the diagnosis because cause of death information was not available. Hospitalisations were included only if COVID-19 was registered in the electronic patient records as the diagnosis for the hospitalisation. The specialty care hospitalisation period was cut to last a maximum of 60 days to exclude hospitalisation periods that may have accidentally not been discontinued in the electronic patient records. Intensive care treatment episodes are included in specialty care episodes. The summary level data were stratified according to predefined groups based on two different criteria:

- the patient's age at the time of diagnosis, and for the hospitalised patients at the start date of hospitalisation (groups: 18–59, 60+, and all ≥18)

- risk groups based on diagnosis/procedure codes registered during a health care visit from 1.1.2010 until the COVID-19 infection (except for cancer, where the requirement was a healthcare visit with a cancer diagnosis code from 1.1.2020 until the COVID-19 infection, to enrich for newly incident cases with active cancer). The risk groups were defined based on previously published data and the list of risk groups for severe COVID-19 published by the Finnish Institute of Health and welfare [10], and included diabetes, cardiovascular (CV) diseases, hypertension, chronic lung disease, organ or stem cell transplant, cancer, chronic kidney disease (CKD), neurological disorder/disease, group with one or more of the above-mentioned conditions, no risk group (patients without any of the conditions mentioned above), and all patients. The risk groups are not mutually exclusive: a patient can belong to multiple risk groups. A table of the risk groups and how they were identified (ICD-10 code or procedure code) is presented in the supplementary material (S3 Table).

## Data validation

Data received from THL for this study was validated using publicly available data from the registry holder THL [11]. Validation of the number of infections was done separately for the age groups included in this study. Validation of deaths was done for the full patient population (S1 Fig).

## Data analysis

Data analysis, including statistical analyses were conducted with R version 4.2.1 (R Foundation for Statistical Computing, Vienna, Austria). Data was analysed through descriptive data analysis (e.g., bar graph visualisations of absolute values and percentages) and comparisons between and inside the groups defined above. The number of people at risk in each age group was defined as the number of inhabitants in Finland in the corresponding age group at the end of 2021 (from Statistics Finland). The number at risk for each of the risk groups included in this study was defined as patients found in the same registers with the same diagnosis or procedure codes as described above for COVID-19 patients, but regardless of their COVID-19 status. The infection hospitalisation rate (IHR) is the proportion of infected individuals who require hospitalisation, and it is used to assess the severity of a disease outbreak, and the case fatality rate (CFR) is the proportion of deaths among individuals diagnosed with a particular disease, often used as a measure of disease severity or the effectiveness of treatments." The IHR and CFR were calculated for each risk group for 1) all adult patients, and for 2) all adult patients under 60 years of age and 3) all patients over 60 years of age. The IHR and CFR were calculated as the number of individuals who were hospitalised due to a COVID-19 infection or died within 30 days of a COVID-19 infection, respectively, divided by the total number of

individuals with a COVID-19 infection. The IHR and CFR were calculated for each risk group, the group with at least one risk group diagnosis, the group without any risk diagnoses, and for the entire population separately. A two proportions z-test was used to test the difference between groups in IHR and in CFR. The statistical significance was set to P = .05. We also analysed the average length of stay (LOS) in both primary and specialty care for patients who were hospitalised with COVID-19 for each risk group.

## Results

Table 1 depicts the size of the patient population and the risk groups that were analysed in this study as well as the number of hospitalisations and deaths in the whole population and the different risk groups. The number of infections has increased over time for all age groups and risk groups, with the highest number of infections reported in H1 2022, when the omicron variant became dominant in Finland.

### Hospitalisations due to COVID-19

The likelihood of requiring inpatient care due to COVID-19 infection by risk group is presented in Table 1 and in Figs 1 (primary care) and 2 (specialty care). In primary care and in specialty care, patients over 60 years old were more likely to be hospitalised than patients under 60 years old during all of the time periods included in the study. Overall, the IHR has decreased over time in both primary and specialty care. There was a statistically significant difference in the IHR between almost all risk groups when compared to the group with no risk factors (S4 Table).

During the most recent time period included in the study (H1 2022), the IHR in the age group 18–59 years old was highest for patients with CKD (10% for specialty care), followed by patients with transplants, cancer, and CV diseases (all 3% for specialty care, Fig 2B). During 2021, hypertension patients also had a relatively high IHR in this age group (specialty care IHRs of 14% and 6% for H1 2021 and H2 2021, respectively, compared to an IHR of 2% for H2 2022, Fig 2B). Notably, almost all COVID-19 related hospitalisations in this age group occurred in specialty care (Figs 1B and 2B).

For the older age group (60+), the primary care IHR in H1 2022 was the highest for patients with CKD (11%), followed by transplant patients, patients with CV diseases, and patients with neurological diseases (9%, 8%, and 8%, respectively, Fig 1C). The specialty care IHRs for this age group during the same time period were the highest for transplant patients (20%), followed by patients with CKD, cancer, and diabetes (18%, 12%, and 11%, respectively, Fig 2C). Overall, about one third of the hospitalisation in this age group occurred in primary care hospitals, and about two thirds in specialty care hospitals. The group with no risk factors had the lowest IHR in all patients ≥18 during all of the time periods included in the study.

### COVID-19 related deaths

The COVID-19 related deaths and case fatality rate (CFR) are presented in Table 1 and Fig 4. Statistical differences are depicted in supplementary material (S4 Table). Overall, the CFR has declined over time. The group with the highest risk of death in the 18+ population at all time periods were patients with CKD (24%, 14%, and 9%, in H1 2021, H2 2021, and H1 2022, respectively, Fig 3A) followed by patients with CV diseases, cancer, and hypertension (5%, 4%, and 4%, respectively in H1 2022, Fig 3A). Additionally, a higher CFR can also be seen in the older age group (60+) with transplant patients (9% in H1 2022, Fig 3C). In general, CFR was greatly affected by age–patients under 60 years old had a maximum CFR of 3% (cancer patients in H1 2021) and patients over 60 years old had CFRs up to 33% (CKD patients in H1

**Table 1. COVID-19 patient population, stratified by age group, risk group and time period.**

| | Age group | Number at risk | Infections (% of number at risk) | | | Hospitalisations (primary care; % of infections) | | | Hospitalisations (specialty care; % of infections) | | | Deaths (% of infections) | | |
|---|---|---|---|---|---|---|---|---|---|---|---|---|---|---|
| | | | -21 H1 | -21 H2 | -22 H1 | -21 H1 | -21 H2 | -22 H1 | -21 H1 | -21 H2 | -22 H1 | -21 H1 | -21 H2 | -22 H1 |
| **All** | 18+ | 4 512 724 | 40 802 (0,9%) | 122 500 (2,7%) | 723 503 (16,0%) | 751 (1,8%) | 1 038 (0,8%) | 3 976 (0,5%) | 2 269 (5,6%) | 2 993 (2,4%) | 8 768 (1,2%) | 431 (1,1%) | 649 (0,5%) | 3362 (0,5%) |
| | 18–59 | 2 878 906 | 35 441 (1,2%) | 107 226 (3,7%) | 623 337 (21,7%) | 110 (0,3%) | 141 (0,1%) | 248 (0,0%) | 1 250 (3,5%) | 1 666 (1,6%) | 2 865 (0,5%) | 31 (0,1%) | 51 (0,0%) | 159 (0,0%) |
| | 60+ | 1 633 818 | 5 361 (0,2%) | 15 274 (0,9%) | 100 166 (6,1%) | 614 (11,5%) | 897 (5,9%) | 3 729 (3,7%) | 1 019 (19,0%) | 1 327 (8,7%) | 5 905 (5,9%) | 400 (7,5%) | 595 (3,9%) | 3 203 (3,2%) |
| **No risk factors** | 18+ | 4 336 420 | 35 022 (0,8%) | 105 233 (2,4%) | 599 833 (13,8%) | 215 (0,6%) | 258 (0,2%) | 783 (0,1%) | 1 330 (3,8%) | 1 694 (1,6%) | 3 159 (0,5%) | 86 (0,2%) | 136 (0,1%) | 568 (0,1%) |
| | 18–59 | 2 783 573 | 32 187 (1,2%) | 97 152 (3,5%) | 552 873 (19,9%) | 68 (0,2%) | 96 (0,1%) | 113 (0,0%) | 940 (2,9%) | 1 268 (1,3%) | 1 814 (0,3%) | 15 (0,0%) | 30 (0,0%) | 65 (0,0%) |
| | 60+ | 1 552 847 | 2 835 (0,2%) | 8 081 (0,5%) | 26 960 (1,7%) | 147 (5,2%) | 162 (2,0%) | 625 (2,3%) | 390 (13,8%) | 426 (5,3%) | 1 346 (5,0%) | 71 (2,5%) | 106 (1,3%) | 503 (1,9%) |
| **≥1 risk factors** | 18+ | 176 304 | 5 780 (3,3%) | 17 267 (9,8%) | 123 670 (70,1%) | 536 (9,3%) | 780 (4,5%) | 3 238 (2,6%) | 939 (16,2%) | 1 299 (7,5%) | 5 608 (4,5%) | 345 (6,0%) | 513 (3,0%) | 2 794 (2,3%) |
| | 18–59 | 95 333 | 3 254 (3,4%) | 10 074 (10,6%) | 70 464 (73,9%) | 42 (1,3%) | 45 (0,4%) | 135 (0,2%) | 310 (9,5%) | 398 (4,0%) | 1 051 (1,5%) | 16 (0,5%) | 24 (0,2%) | 94 (0,1%) |
| | 60+ | 80 971 | 2 526 (3,1%) | 7 193 (8,9%) | 53 206 (65,7%) | 494 (19,6%) | 735 (10,2%) | 3 104 (5,8%) | 629 (24,9%) | 901 (12,5%) | 4 559 (8,6%) | 329 (13,0%) | 489 (6,8%) | 2 700 (5,1%) |
| **Cancer** | 18+ | 27 572 | 744 (2,7%) | 2 315 (8,4%) | 18 390 (66,7%) | 141 (19,0%) | 167 (7,2%) | 791 (4,3%) | 144 (19,4%) | 254 (11,0%) | 1 486 (8,1%) | 97 (13,0%) | 124 (5,4%) | 707 (3,8%) |
| | 18–59 | 10 575 | 280 (2,6%) | 935 (8,8%) | 7 507 (71,0%) | 7 (2,5%) | 7 (0,7%) | 21 (0,3%) | 26 (9,3%) | 55 (5,9%) | 211 (2,8%) | 9 (3,2%) | <5 (0,4%) | 38 (0,5%) |
| | 60+ | 16 997 | 464 (2,7%) | 1 380 (8,1%) | 10 883 (64,0%) | 134 (28,9%) | 160 (11,6%) | 770 (7,1%) | 118 (25,4%) | 199 (14,4%) | 1 276 (11,7%) | 88 (19,0%) | 120 (8,7%) | 669 (6,1%) |
| **Chronic lung disease** | 18+ | 51 598 | 1 635 (3,2%) | 5 270 (10,2%) | 37 515 (72,7%) | 127 (7,8%) | 215 (4,1%) | 789 (2,1%) | 265 (16,2%) | 379 (7,2%) | 1 474 (3,9%) | 95 (5,8%) | 112 (2,1%) | 595 (1,6%) |
| | 18–59 | 35 184 | 1 127 (3,2%) | 3 808 (10,8%) | 26 884 (76,4%) | 10 (0,9%) | 14 (0,4%) | 44 (0,2%) | 97 (8,6%) | 112 (2,9%) | 300 (1,1%) | <5 (0,1%) | 8 (0,2%) | 17 (0,1%) |
| | 60+ | 16 414 | 508 (3,1%) | 1 462 (8,9%) | 10 631 (64,8%) | 117 (23,0%) | 201 (13,7%) | 746 (7,0%) | 168 (33,1%) | 267 (18,3%) | 1 174 (11,0%) | 94 (18,5%) | 104 (7,1%) | 578 (5,4%) |
| **CKD[a]** | 18+ | 6 185 | 218 (3,5%) | 536 (8,7%) | 4 144 (67,0%) | 51 (23,4%) | 89 (16,6%) | 338 (8,2%) | 84 (38,5%) | 125 (23,3%) | 655 (15,8%) | 52 (23,9%) | 75 (14,0%) | 362 (8,7%) |
| | 18–59 | 1 682 | 64 (3,8%) | 177 (10,5%) | 1 177 (70,0%) | <5 (6,3%) | <5 (1,1%) | 9 (0,8%) | 21 (32,8%) | 28 (15,8%) | 119 (10,1%) | <5 (1,6%) | <5 (1,7%) | 7 (0,6%) |
| | 60+ | 4 503 | 154 (3,4%) | 359 (8,0%) | 2 967 (65,9%) | 47 (30,5%) | 87 (24,2%) | 330 (11,1%) | 63 (40,9%) | 97 (27,0%) | 536 (18,1%) | 51 (33,1%) | 72 (20,1%) | 355 (12,0%) |
| **CV diseases[b]** | 18+ | 53 286 | 1 617 (3,0%) | 4 722 (8,9%) | 35 973 (67,5%) | 301 (18,6%) | 484 (10,2%) | 1 981 (5,5%) | 350 (21,6%) | 583 (12,3%) | 2 978 (8,3%) | 201 (12,4%) | 346 (7,3%) | 1 782 (5,0%) |
| | 18–59 | 14 890 | 498 (3,3%) | 1 583 (10,6%) | 10 667 (71,6%) | 9 (1,8%) | 19 (1,2%) | 40 (0,4%) | 67 (13,5%) | 95 (6,0%) | 296 (2,8%) | 6 (1,2%) | 10 (0,6%) | 29 (0,3%) |
| | 60+ | 38 396 | 1 119 (2,9%) | 3 139 (8,2%) | 25 306 (65,9%) | 292 (26,1%) | 465 (14,8%) | 1 942 (7,7%) | 293 (26,2%) | 488 (15,5%) | 2 683 (10,6%) | 195 (17,4%) | 336 (10,7%) | 1 753 (6,9%) |
| **Diabetes** | 18+ | 36 851 | 1 269 (3,4%) | 3 339 (9,1%) | 25 266 (68,5%) | 149 (11,7%) | 245 (7,3%) | 999 (4,0%) | 265 (20,9%) | 406 (12,2%) | 1 775 (7,0%) | 121 (9,5%) | 164 (4,9%) | 872 (3,5%) |
| | 18–59 | 16 584 | 609 (3,7%) | 1 610 (9,1%) | 11 879 (71,6%) | 8 (1,3%) | 10 (0,6%) | 36 (0,3%) | 78 (12,8%) | 104 (6,5%) | 295 (2,5%) | <5 (0,7%) | 8 (0,5%) | 21 (0,2%) |
| | 60+ | 20 267 | 660 (3,3%) | 1 729 (8,5%) | 13 347 (65,9%) | 141 (21,4%) | 235 (13,6%) | 963 (7,2%) | 187 (28,3%) | 302 (17,5%) | 1480 (11,1%) | 117 (17,5%) | 156 (9,0%) | 851 (6,4%) |

(*Continued*)

**Table 1.** (Continued)

| | Age group | Number at risk | Infection(s) (% of number at risk) | | | Hospitalisations (primary care; % of infections) | | | Hospitalisations (specialty care; % of infections) | | | Deaths (% of infections) | | |
|---|---|---|---|---|---|---|---|---|---|---|---|---|---|---|
| **Hypertension** | 18+ | 78 987 | 4 133 (5,2%) | 11 797 (14,9%) | 89 516 (113,3%) | 370 (9,0%) | 527 (4,5%) | 2 265 (2,5%) | 529 (12,8%) | 746 (6,3%) | 3 463 (3,9%) | 226 (5,5%) | 335 (2,8%) | 1 950 (2,2%) |
| | 18–59 | 26 873 | 881 (3,3%) | 2 518 (9,4%) | 19 158 (71,3%) | 12 (1,4%) | 20 (0,8%) | 67 (0,3%) | 127 (14,4%) | 156 (6,2%) | 405 (2,1%) | 6 (0,7%) | 16 (0,6%) | 43 (0,2%) |
| | 60+ | 52 114 | 1 615 (3,1%) | 4 494 (8,6%) | 34 167 (65,6%) | 358 (22,2%) | 507 (11,3%) | 2 198 (6,4%) | 402 (24,9%) | 590 (13,1%) | 3 060 (9,0%) | 220 (13,6%) | 319 (7,1%) | 1 907 (5,6%) |
| **Neurological disorder / disease** | 18+ | 6 828 | 177 (2,6%) | 601 (8,8%) | 4 822 (70,6%) | 28 (15,8%) | 40 (6,7%) | 166 (3,4%) | 25 (14,1%) | 42 (7,0%) | 235 (4,9%) | 20 (11,3%) | 29 (4,8%) | 159 (3,3%) |
| | 18–59 | 3 804 | 107 (2,8%) | 376 (9,9%) | 2 791 (73,4%) | <5 (2,8%) | <5 (0,3%) | 7 (0,3%) | 11 (10,3%) | 16 (4,3%) | 42 (1,5%) | <5 (0,0%) | <5 (0,5%) | 5 (0,2%) |
| | 60+ | 3 024 | 70 (2,3%) | 225 (7,4%) | 2 031 (67,2%) | 25 (35,7%) | 39 (17,3%) | 159 (7,8%) | 14 (20,0%) | 26 (11,6%) | 193 (9,5%) | 20 (28,6%) | 27 (12,0%) | 154 (7,6%) |
| **Organ or stem cell transplant** | 18+ | 11 569 | 508 (4,4%) | 1 409 (12,2%) | 8 164 (70,6%) | 34 (6,7%) | 47 (3,3%) | 201 (2,5%) | 81 (15,9%) | 104 (7,4%) | 631 (7,7%) | 27 (5,3%) | 40 (2,8%) | 217 (2,7%) |
| | 18–59 | 8 272 | 403 (4,9%) | 1 146 (13,9%) | 5 997 (72,5%) | 8 (2,0%) | <5 (0,3%) | 12 (0,2%) | 36 (8,9%) | 47 (4,1%) | 206 (3,4%) | <5 (0,2%) | <5 (0,3%) | 18 (0,3%) |
| | 60+ | 3 297 | 105 (3,2%) | 263 (8,0%) | 2 167 (65,7%) | 26 (24,8%) | 44 (16,7%) | 190 (8,8%) | 45 (42,9%) | 57 (21,7%) | 426 (19,7%) | 26 (24,8%) | 37 (14,1%) | 199 (9,2%) |

[a]CKD = chronic kidney disease

[b]CV = cardiovascular

2021), depending on the risk group and time period. The group with no risk factors had the lowest CFR in all patients ≥18 during all of the time periods included in the study.

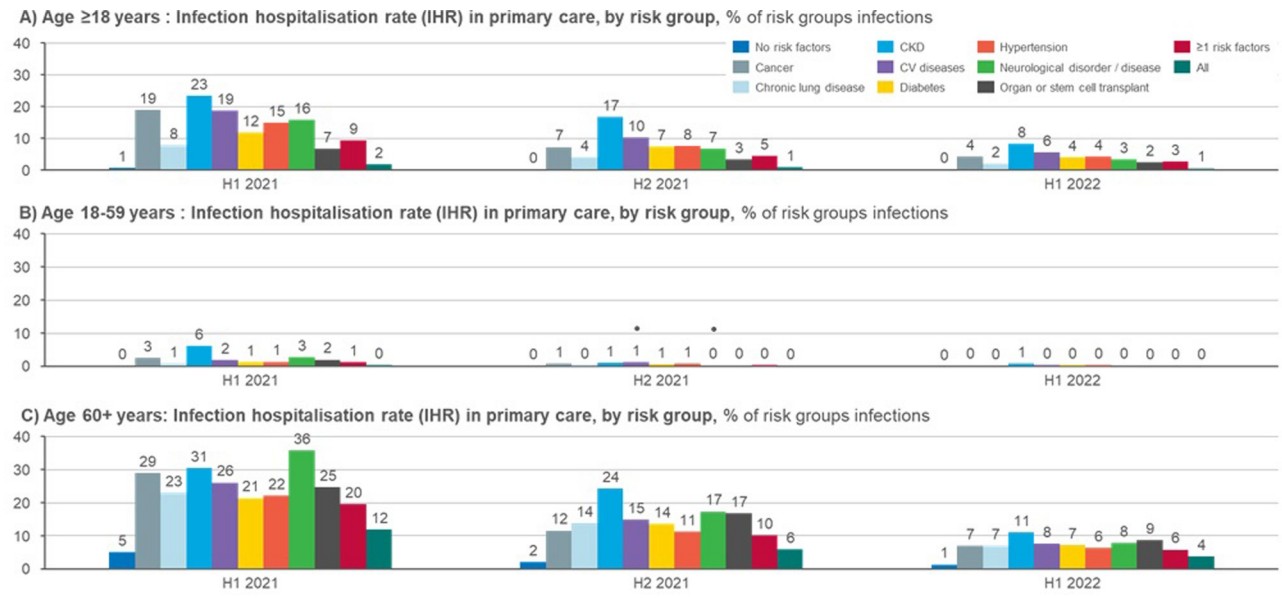

**Fig 1. Infection hospitalisation rate (IHR; % of risk groups infections) in primary care, by risk group and age group.** Non-significant differences (between risk group and group with none of the specified risk factors) are marked with ●.

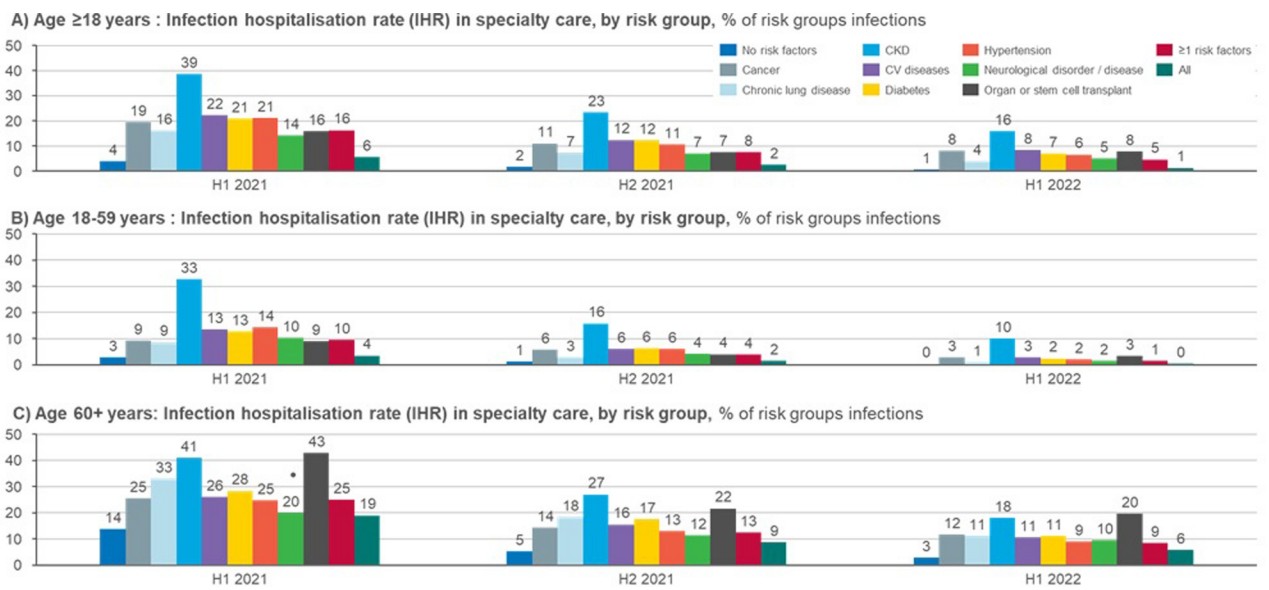

**Fig 2. Infection hospitalisation rate (IHR; % of risk groups infections) in specialty care, by risk group and age group.** Non-significant differences (between risk group and group with none of the specified risk factors) are marked with ●.

## Length of hospital stay

The average LOS is presented in Fig 4 for all hospitalised patients. In primary care (Fig 4A–4C), the mean LOS for all hospitalised patients ≥18 years ranged from 15 to 27 days depending on the risk group, and in specialty care, from 7 to 17 days depending on the risk group. In specialty care the LOS has decreased during the study period for all risk groups when comparing H1 2021 to H1 2022. The biggest change can be seen in patients with neurological diseases, where the LOS in specialty care has decreased from 15 to 7 (H1 2021 to H1 2022, respectively).

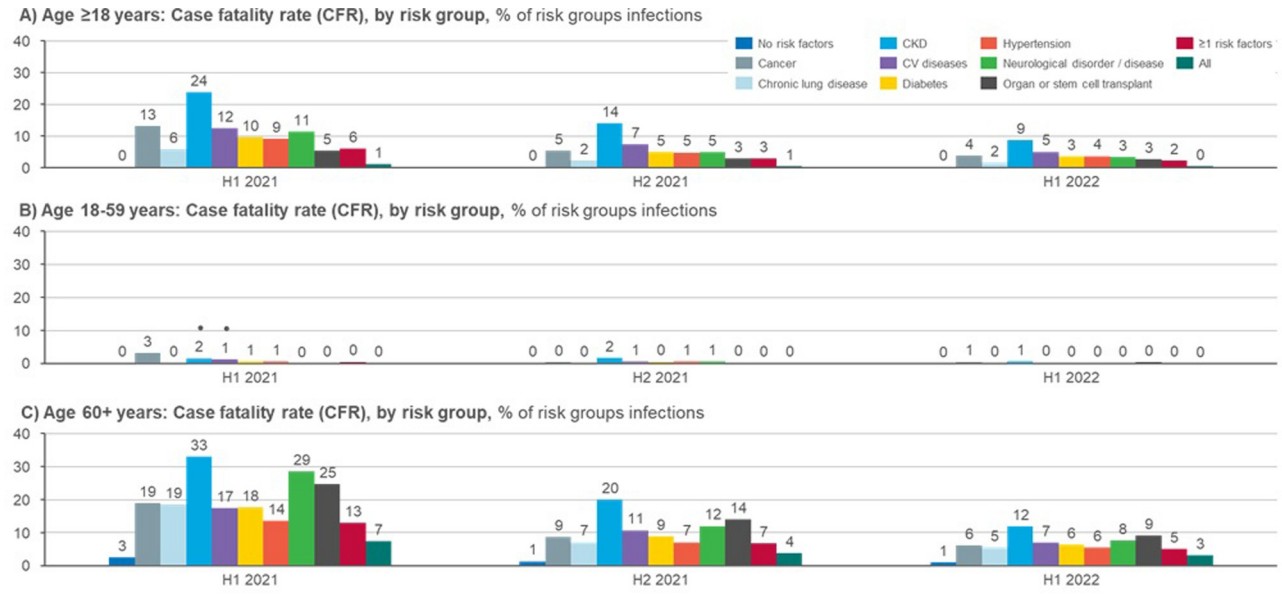

**Fig 3. Case fatality rate (CFR; % of risk groups infections), by risk group and age group.** Non-significant differences (between risk group and group with none of the specified risk factors) are marked with ●.

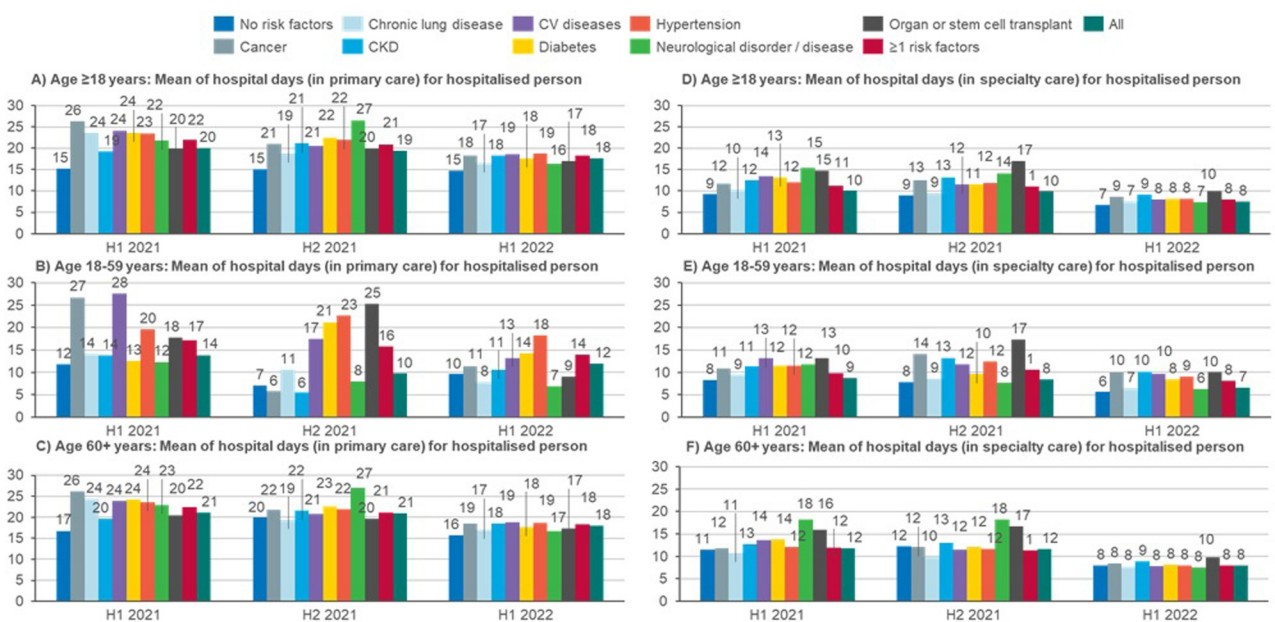

**Fig 4.** The length of stay (LOS) in primary care (A-C) and in specialty care (D-F) hospitals due to COVID-19 infection, by risk group and age.

The group with no risk factors has the lowest average LOS in all hospitalised patients ≥18 during all of the studied time periods in both primary and specialty care.

## Discussion

In recent studies from different Nordic countries, differences between the delta-dominant and omicron-dominant periods have been estimated for example on risk of hospitalisation associated with infection [12], risk of hospitalisation, death, and length of stay [13], and on risk of severe COVID-19 in relation to vaccination status, sex, age, and comorbidities [14]. To our knowledge, this is the first study from the Nordic countries showing which risk group patients have the highest risk of hospitalisation or death due to COVID-19 during both the delta-dominant and the omicron-dominant periods of the COVID-19 pandemic. Our study shows that patients belonging to one of the risk groups defined in this study, such as CKD, cancer, or transplant patients, have a higher risk of hospitalisation and death after a COVID-19 infection, compared to patients without these risk factors. Our study also confirms the previously published findings [6, 8] that older age is a significant risk factor for severe COVID-19, and that the risk is further increased in people with comorbidities. Elderly patients (60+) are relatively more often hospitalised in primary care as the general condition of the elderly is often poorer and requires medical care and monitoring while younger patients are more likely able to recover in a similar situation at home.

Our study indicates that both the infection hospitalization rate (IHR) and the case fatality rate (CFR) decreased during the study period in all studied age and risk groups. This is in line with findings from previous studies [13]. Also, the mean length of stay (LOS) for patients admitted to the hospital decreased during the study period for both primary care and secondary care hospitalisations.

The changes over time in all evaluated parameters may be partially explained by changes in testing, hospitalisation guidelines, vaccination coverage, COVID-19 medications, and differences in disease severity caused by the different COVID-19 variants. Testing recommendations

and possibilities have changed several times during the study period, e.g., in May 2021 home-testing became possible in Finland and from September 2021 onwards official (PCR) testing has been focused on people who are unvaccinated or have an increased risk of a severe COVID-19 infection [15]. However, the changes in the testing strategy towards high-risk groups should increase the IHR and CFR, since a larger share of laboratory confirmed cases are among the risk groups. Thus, the changes in the testing policy do not explain the observed trends.

Furthermore, vaccination coverage has increased during the study period (S5 Table). In Finland, vaccinations began in early 2021 with the 80+ population and other high-risk groups, and the coverage reached over 90% for the 60+ group by the end of H1 2021 for the first dose, 90% for the second dose during H2 2021, and 70% by the end of H1 2022 for third dose [16]. Each new round of vaccinations started with high-risk groups including the elderly population. The vaccination coverage for the younger age group (18–59 years) increased to over 80% during H2 2021 for the first and second dose and to almost 60% by the end of H1 2022 for the third dose [16]. There are no publicly available statistics on the vaccination coverage of individual risk groups, but as people with a high risk for severe disease have been offered vaccines earlier than the general population, their vaccination coverage can be expected to exceed that of the general population. The second dose administered to high-risk groups in the summer of 2021 most likely contributed to the decreasing trends in IHR and CFR. The trends can also be partially explained by the less severe nature of the omicron variant, which became dominant towards the end of December 2021 in Finland [17].

Due to the multiple changes in COVID-19 variants and the healthcare environment that took place during the 18-month study period, the comparisons between risk groups are more reliable than comparisons over time. Additionally, since the hospital admission criteria did not change significantly during the observation period, the LOS can be compared between time points as well as between risk groups. The observed decrease in the overall LOS during the study period can likely be explained by less severe disease caused by the omicron-variant, as well as increased vaccination coverage.

## Study limitations

The data from 2022 has not been validated by THL and there can be minor corrections to it during the validation process. Usually, these corrections have the same effect on all risk groups. However, they might lower the reliability of comparisons between H1-H2 2021 and H1 2022. Another limitation of the study is that due to diagnosis recording practises the data may contain a small number of inpatient episodes for patients who were hospitalised primarily for other reasons but who happened to test positive for COVID-19. However, this should increase the IHR rather than decrease it. Furthermore, because deaths were included if they occurred within 30 days of a verified infection, it is possible that deaths due to reasons other than COVID-19 were included, and COVID-19 deaths that occurred later than 30 days after an infection were excluded. We have mitigated these limitations through validating data with the public registry resources that are available (S1 Fig). Due to the limitations of the study, the comparisons between different risk groups are more reliable than the absolute values of the variables. The number of patients under 60 years old in primary care are small, therefore all statistically significant differences between IHRs need to be interpreted with caution. There might also be some bias on risk group hospitalisations compared to the general population, as clinicians might admit risk group patients to the hospital with a lower threshold.

Data for this study was collected on an aggregate level, which means that statistical testing is not possible to the same extent as with patient level data.

The risk groups have been formed based on ICD-10-codes and procedure codes. Some known risk factors for severe COVID-19, such as obesity and smoking, are not reliably recorded in registry data, so these risk factors are not included in our study.

COVID-19 vaccinations are recorded in a different registry, and therefore data on patient level vaccination status is missing. This means that the effect of vaccinations on IHR, CFR or length of stay could not be taken into consideration in the analysis. However, the vaccination coverage was high during the study period, and therefore, our results depict mainly the risk of vaccinated patients, especially in the 60+ age group in H2 2021 and H1 2022.

## Conclusion

Despite the decrease in COVID-19 hospitalisations and deaths observed during the study period, a significant proportion of patients are still hospitalised, and deaths occur especially in the 60+ population. Also, even though the average length of stay of hospitalised COVID-19 patients has decreased over the study period, it is still long compared to specialty care hospitalisations in general, with an average of four days in Finland [18]. Old age is a significant risk factor for severe COVID-19 in all patient groups, seen especially in CFR. In addition to age, certain risk factors such as CKD clearly increase the risk for severe COVID-19 outcomes. Early treatment should be considered with a low threshold for risk group patients and for elderly patients in order to avoid severe disease courses, and to ease the burden on hospitals where resources are currently very strained. Further studies are needed to investigate the impact of early treatment options for COVID-19 on hospitalisations and deaths.

## Supporting information

**S1 Fig. Data validation example: Number of deaths in the study population compared to THL public data.**
(TIF)

**S1 Table. Difference between the number of patients with a COVID-19 infection identified from THL registers (Hilmo and Avohilmo) and patients identified only in the TTR (primary and specialty care separately) (n = THL-TTR, % = 1-(THL-TTR)/THL).**
(PDF)

**S2 Table. Difference between days in hospital care for patients identified from THL registers (Hilmo and Avohilmo) and for patients identified only in the TTR (primary and specialty care separately) (n = THL-TTR, % = 1-(THL-TTR)/THL).**
(PDF)

**S3 Table. Risk groups and identification of risk groups from data.**
(PDF)

**S4 Table. P-values of statistical testing for IHR and CFR.** All groups are compared to the no risk -group.
(PDF)

**S5 Table. COVID-19 vaccination coverage in Finland (n, % of population).**
(PDF)

**S1 File. Data.**
(XLSX)

## Author Contributions

**Conceptualization:** Milla Summanen, Mikko Kosunen, Ville Kainu, Severi Niskanen, Outi Isomeri.

**Formal analysis:** Severi Niskanen, Lalli Nurmi.

**Funding acquisition:** Milla Summanen.

**Investigation:** Anniina Cansel, Lalli Nurmi.

**Methodology:** Mikko Kosunen, Anniina Cansel, Severi Niskanen, Lalli Nurmi, Outi Isomeri.

**Project administration:** Milla Summanen, Mikko Kosunen, Outi Isomeri.

**Supervision:** Milla Summanen, Riikka-Leena Leskelä, Outi Isomeri.

**Validation:** Mikko Kosunen, Anniina Cansel, Severi Niskanen, Lalli Nurmi.

**Visualization:** Anniina Cansel, Lalli Nurmi.

**Writing – original draft:** Milla Summanen, Mikko Kosunen, Ville Kainu, Anniina Cansel, Severi Niskanen, Riikka-Leena Leskelä, Outi Isomeri.

**Writing – review & editing:** Milla Summanen, Mikko Kosunen, Ville Kainu, Anniina Cansel, Severi Niskanen, Lalli Nurmi, Riikka-Leena Leskelä, Outi Isomeri.

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
