## [Decision Letter · Decision Letter 0]

20 Mar 2023

PONE-D-23-03744Covid-19 hospitalisations and all-cause mortality by risk group in FinlandPLOS ONE

Dear Dr. Cansel,

Thank you for submitting your manuscript to PLOS ONE. After careful consideration, we feel that it has merit but does not fully meet PLOS ONE’s publication criteria as it currently stands. Therefore, we invite you to submit a revised version of the manuscript that addresses the points raised during the review process.

We look forward to receiving your revised manuscript.

Kind regards,

K M Amran Hossain, MScPT

Academic Editor

PLOS ONE

Journal Requirements:

2. You indicated that ethical approval was not necessary for your study. We understand that the framework for ethical oversight requirements for studies of this type may differ depending on the setting and we would appreciate some further clarification regarding your research. Could you please provide further details on why your study is exempt from the need for approval and confirmation from your institutional review board or research ethics committee (e.g., in the form of a letter or email correspondence) that ethics review was not necessary for this study? Please include a copy of the correspondence as an ""Other"" file.

   "The study was funded by Pfizer Oy. AC, SN, LN, R-LL and OI are employees of Nordic Healthcare Group, which received funding from Pfizer Oy in connection with the development of this manuscript."

Additional Editor Comments:

Dear Authors

Reviewers suggested to revise your manuscript, if you wish to proceed, we may proceed to the final acceptance to your paper.

If you agree, please proceed the revision addressing the reviewer's comments

Reviewers' comments:

Reviewer's Responses to Questions

**Comments to the Author**

1. Is the manuscript technically sound, and do the data support the conclusions?

Reviewer #1: Yes

Reviewer #2: Yes

2. Has the statistical analysis been performed appropriately and rigorously? 

Reviewer #1: Yes

Reviewer #2: Yes

3. Have the authors made all data underlying the findings in their manuscript fully available?

Reviewer #1: Yes

Reviewer #2: Yes

4. Is the manuscript presented in an intelligible fashion and written in standard English?

Reviewer #1: Yes

Reviewer #2: Yes

5. Review Comments to the Author

Reviewer #1: First, I appreciate the authors efforts in conducting and presenting this interesting study. I understand that the peer review process can be challenging, but please rest assured that my aim is to provide constructive feedback that will help to improve the quality of your manuscript and contribute to its overall success.

1. Please use the term "COVID-19" instead of "Covid-19".

2. Abstract: The abstract presents a clear and concise overview of a retrospective observational study on COVID-19 in Finland, including the study's objectives, methods, and key findings. The abstract is written in a scientific style, and the language used is appropriate for a scientific audience. Overall, the abstract appears to be well-written and informative.

3. Introduction: Some minor improvements can be suggested as follows-

a) In the line 42, it should be noted that the World Health Organisation (WHO) announced COVID-19 as a global pandemic, not just "announced" it.

b) In the lines 44 and 45, it should be noted that while most people infected with COVID-19 have mild-to-moderate symptoms, a significant proportion may still experience severe symptoms leading to hospitalization and death.

c) In the line 46, it should be noted that while certain risk groups are more likely to develop severe COVID-19, the disease can affect anyone regardless of age or pre-existing conditions.

d) In the line 47, please add Chronic respiratory diseases as pre-existing condition, you may also add diabetes, because people with diabetes s and COVID-19 often need invasive ventilation care and need intensive care unit (ICU) due to their likelihood of developing Acute Respiratory Distress Syndrome (ARDS) -Aging (Albany NY). 2020 Apr 15; 12(7): 6049–6057. Published online 2020 Apr 8. doi: 10.18632/aging.103000)

e) In the line 53 and 54, it should be noted that while COVID-19 vaccines can reduce the burden of the disease, they do not completely eliminate the risk of severe symptoms or transmission.

f) In the line 56 and 57, it should be noted that while early treatment may prevent severe COVID-19, it is not always effective and should be used in combination with preventive measures and vaccination.

(Overall, the introduction provides a reasonable and accurate overview of the current understanding of COVID-19 and its risk factors, while acknowledging some uncertainties and the importance of ongoing research.)

4. Methods: There are no major scientific writing errors in the methods. However, here are a few minor suggestions for improvement-

a) In the line 72, the phrase "summary level statistics" could be clarified for readers who may not be familiar with the term. For example, "This study used aggregate data obtained from the Finnish Institute for Health and Welfare's registers..."

b) In the lines 83, 84 and 85, "The results are analysed based on patients identified from Hilmo..." could be rephrased for clarity, for example: "Data from Hilmo was used to identify patients for analysis, and any differences between TTR and Hilmo patients are reported in supplementary materials (S1 and S2 Tables)."

c) In the lines 113 and 114, it may be helpful to specify which types of visualizations were used for the descriptive data analysis.

d) In the line 119, it may be helpful to provide a brief explanation of the infection hospitalization rate (IHR) and the case fatality rate (CFR), for readers who may not be familiar with these terms.

e) In the lines 122 and 123, the phrase "divided by the total number of individuals with a COVID-19 infection" could be clarified by specifying whether this refers to the total number of infected individuals in the entire population or only within each risk group.

5. Results: Well written and described

6. Discussion:

a) In the lines 201 to 205, elderly with COVID-19 infection was only indentified as a risk group in this study? If not, please add some other risk factors points like Chronic lung diseases, diabetes, cardiovascular diseases. My suggestion stands: https://doi.org/10.1371/journal.pone.0233147;
https://doi.org/10.1016/S2213-2600(20)30116-8.

Reviewer #2: REVIEWERS COMMENTS- PONE-D-23-03744

The study PONE-D-23-03744: Covid-19 hospitalizations and all-cause mortality by risk group” in Finland” aimed to identify which risk groups are at the highest risk of having

severe Covid-19 infection, sever disease leading to hospitalization or death in Finland. The authors used retrospective observational study with Finnish registry data to answer this question.

On the whole, the manuscript is well written, clearly followed and easily understood.

Below are a few issues for authors consideration;

Background

Line 61: why Omicron in particular?

Materials and methods

Line 76-77: was the 2020 data unavailable especially the later months of the year

Discussions

Line 197-199: please cross-check this claim to be sure- there seems to many studies in that region reporting covid-19 severities and related mortalities vs risk factors

Line 201- 205:

6. PLOS authors have the option to publish the peer review history of their article (what does this mean?). If published, this will include your full peer review and any attached files.

Reviewer #1: No

Reviewer #2: No

---

## [Author Response · Author response to Decision Letter 0]

1 May 2023

PLOS ONE’s style requirements: The PLOS ONE’s style requirements have been reviewed and needed changes have been made into the manuscript and files.

Changes to financial disclosure: The study was funded by Pfizer Oy. AC, SN, LN, R-LL and OI are employees of Nordic Healthcare Group, which received funding from Pfizer Oy in connection with the development of this manuscript. Role of Funder statement: The funders participated in study design, analysis, decision to publish and preparation of the manuscript. The funders had no role in data collection. 

Changes to data availability statement: The minimal data set is provided in the Supporting Information files as aggregate level data. Patient-level data is only available from register holder. 

Clarification to ethical approval: The ethical approval is not necessary based on the Act on the Secondary Use of Health and Social Data in Finland.

Changes in the references: Reference list has been reviewed and some additions (according to the recommendations of the reviewers) have been made (e.g., doi numbers have been added and the webpages have been reviewed).

Response to reviewers

The authors’ comments can be found in red below.

5. Review Comments to the Author

Reviewer #1: First, I appreciate the authors efforts in conducting and presenting this interesting study. I understand that the peer review process can be challenging, but please rest assured that my aim is to provide constructive feedback that will help to improve the quality of your manuscript and contribute to its overall success.

Dear Reviewer,

Thank you very much for your thoughtful review of our manuscript. We greatly appreciate your kind words and your effort to provide constructive feedback that will help to improve the quality of our research. Your comments have been invaluable in shaping the final version of our manuscript, and we are grateful for the time and effort you have invested in this process.

1. Please use the term "COVID-19" instead of "Covid-19". This change has been made throughout the manuscript.

2. Abstract: The abstract presents a clear and concise overview of a retrospective observational study on COVID-19 in Finland, including the study's objectives, methods, and key findings. The abstract is written in a scientific style, and the language used is appropriate for a scientific audience. Overall, the abstract appears to be well-written and informative.

3. Introduction: Some minor improvements can be suggested as follows-

a) In the line 42, it should be noted that the World Health Organisation (WHO) announced COVID-19 as a global pandemic, not just "announced" it. This change has been made to the manuscript.

b) In the lines 44 and 45, it should be noted that while most people infected with COVID-19 have mild-to-moderate symptoms, a significant proportion may still experience severe symptoms leading to hospitalization and death. This change has been made to the manuscript.

c) In the line 46, it should be noted that while certain risk groups are more likely to develop severe COVID-19, the disease can affect anyone regardless of age or pre-existing conditions. This change has been made to the manuscript.

d) In the line 47, please add Chronic respiratory diseases as pre-existing condition, you may also add diabetes, because people with diabetes s and COVID-19 often need invasive ventilation care and need intensive care unit (ICU) due to their likelihood of developing Acute Respiratory Distress Syndrome (ARDS) -Aging (Albany NY). 2020 Apr 15; 12(7): 6049–6057. Published online 2020 Apr 8. doi: 10.18632/aging.103000) This change has been made to the manuscript and the suggested reference has been added.

e) In the line 53 and 54, it should be noted that while COVID-19 vaccines can reduce the burden of the disease, they do not completely eliminate the risk of severe symptoms or transmission. This change has been made to the manuscript.

f) In the line 56 and 57, it should be noted that while early treatment may prevent severe COVID-19, it is not always effective and should be used in combination with preventive measures and vaccination.

(Overall, the introduction provides a reasonable and accurate overview of the current understanding of COVID-19 and its risk factors, while acknowledging some uncertainties and the importance of ongoing research.) This change has been made to the manuscript.

4. Methods: There are no major scientific writing errors in the methods. However, here are a few minor suggestions for improvement-

a) In the line 72, the phrase "summary level statistics" could be clarified for readers who may not be familiar with the term. For example, "This study used aggregate data obtained from the Finnish Institute for Health and Welfare's registers..." This change has been made to the manuscript.

b) In the lines 83, 84 and 85, "The results are analysed based on patients identified from Hilmo..." could be rephrased for clarity, for example: "Data from Hilmo was used to identify patients for analysis, and any differences between TTR and Hilmo patients are reported in supplementary materials (S1 and S2 Tables)." This change has been made to the manuscript.

c) In the lines 113 and 114, it may be helpful to specify which types of visualizations were used for the descriptive data analysis. This change has been made to the manuscript.

d) In the line 119, it may be helpful to provide a brief explanation of the infection hospitalization rate (IHR) and the case fatality rate (CFR), for readers who may not be familiar with these terms. This change has been made to the manuscript.

e) In the lines 122 and 123, the phrase "divided by the total number of individuals with a COVID-19 infection" could be clarified by specifying whether this refers to the total number of infected individuals in the entire population or only within each risk group. This change has been made to the manuscript.

5. Results: Well written and described

6. Discussion:

a) In the lines 201 to 205, elderly with COVID-19 infection was only indentified as a risk group in this study? If not, please add some other risk factors points like Chronic lung diseases, diabetes, cardiovascular diseases. My suggestion stands: https://doi.org/10.1371/journal.pone.0233147;
https://doi.org/10.1016/S2213-2600(20)30116-8. The other risk factors have been discussed now in the sentence earlier to this.

Reviewer #2: REVIEWERS COMMENTS- PONE-D-23-03744

The study PONE-D-23-03744: Covid-19 hospitalizations and all-cause mortality by risk group” in Finland” aimed to identify which risk groups are at the highest risk of having

severe Covid-19 infection, sever disease leading to hospitalization or death in Finland. The authors used retrospective observational study with Finnish registry data to answer this question.

On the whole, the manuscript is well written, clearly followed and easily understood.

Dear Reviewer,

We would like to express our sincere gratitude for your thoughtful review of our manuscript. Your kind words and feedback have been invaluable to us and we greatly appreciate your time and effort in providing us with your helpful comments. Thank you once again for your contribution to our manuscript. We have taken your feedback on board and made the necessary revisions to improve the quality of our work.

Below are a few issues for authors consideration;

Background

Line 61: why Omicron in particular? We wanted to focus on Omicron due to the fact that most of the published research has been about earlier virus variants. Additionally, the disease itself is different than during the earlier variants.

Materials and methods

Line 76-77: was the 2020 data unavailable especially the later months of the year The earlier data had been already reported and published in a research article by THL and therefore we wanted to focus on the newest datasets available (reference 8. Auro, K, Paajanen, T, Koskelainen, S, Vaara, S, Brunfeldt, M, Hannila-Handelberg, T et al. COVID-19-pandemian tunnusluvut Suomessa: ensimmäinen pandemiavuosi. Duodecim, 2022, 138.9: 821-830. Available from: https://www.duodecimlehti.fi/xmedia/duo/duo16741.pdf). Additionally, we wanted to include the time that vaccinations were already available in Finland: the vaccinations started early 2021 and the high-risk groups were prioritised in vaccinations.

Discussions

Line 197-199: please cross-check this claim to be sure- there seems to many studies in that region reporting covid-19 severities and related mortalities vs risk factors This has been cross-checked and to our knowledge only articles including data from 2020 have been published with some of the risk factors that we have also studied. Our study seems to be the first one from the Nordics reporting data from 2021 to 2022, including the omicron-dominant period of the pandemic.

Line 201- 205:

---

## [Editor Report · Decision Letter 1]

10 May 2023

COVID-19 hospitalisations and all-cause mortality by risk group in Finland

PONE-D-23-03744R1

Dear Dr. Cansel,

We’re pleased to inform you that your manuscript has been judged scientifically suitable for publication and will be formally accepted for publication once it meets all outstanding technical requirements.

Kind regards,

K M Amran Hossain, MScPT

Academic Editor

PLOS ONE

Additional Editor Comments (optional):

Thank you for the revision of the manuscript
---

## [Editor Report · Acceptance letter]

15 May 2023

PONE-D-23-03744R1 

COVID-19 hospitalisations and all-cause mortality by risk group in Finland 

Dear Dr. Cansel:

I'm pleased to inform you that your manuscript has been deemed suitable for publication in PLOS ONE. Congratulations! Your manuscript is now with our production department. 

Kind regards, 

on behalf of

Dr. K M Amran Hossain 

Academic Editor

PLOS ONE